# A burst-dependent thalamocortical substrate for perceptual awareness

**Christopher J. Whyte**[1,2], **Eli J. Müller**[1,2], **Jaan Aru**[3], **Matthew Larkum**[4,5], **Yohan John**[6], **Brandon R. Munn**[1,2‡], **James M. Shine**[1,2‡]*

1 Centre for Complex Systems, The University of Sydney, Sydney, New South Wales, Australia, 2 Brain and Mind Center, The University of Sydney, Sydney, New South Wales, Australia, 3 Computational Neuroscience Lab, University of Tartu, Tartu, Estonia, 4 Institute for Biology, Humboldt University of Berlin, Berlin, Germany, 5 NeuroCure Cluster of Excellence, Charité – Universitätsmedizin Berlin, Berlin, Germany, 6 Department of Health Sciences, Neural Systems Laboratory, Boston University, Boston, Massachusetts, United States of America

‡These authors share senior authorship on this work.
*mac.shine@sydney.edu.au

## Abstract

Contemporary models of perceptual awareness lack tractable neurobiological constraints. Inspired by recent cellular recordings in a mouse model of tactile threshold detection, we constructed a biophysical model of perceptual awareness that incorporated essential features of thalamocortical anatomy and cellular physiology. Our model reproduced, and mechanistically explains, the key *in vivo* neural and behavioural signatures of perceptual awareness in the mouse model, as well as the response to a set of causal perturbations. We generalised the same model (with identical parameters) to a more complex task – visual rivalry – and found that the same thalamic-mediated mechanism of perceptual awareness determined perceptual dominance. This led to the generation of a set of novel, and directly testable, electrophysiological predictions. Analyses of the model based on dynamical systems theory show that perceptual awareness in simulations of both threshold detection and visual rivalry arises from the emergent systems-level dynamics of thalamocortical loops.

## Author summary

Computational models of perceptual awareness lack neurobiological grounding. Inspired by recent cellular recordings in a mouse model of tactile threshold detection, we developed a biophysical model of perceptual awareness incorporating key elements of the relevant mesoscale thalamocortical anatomy and cellular physiology. The model reproduced the key neural and behavioural signatures of perceptual awareness in the threshold detection task and the animal's responses to causal perturbations of the thalamocortical circuit. We then generalised the same model to visual rivalry, an experimental paradigm typically used to study perceptual awareness in human and non-human primates. Analysis of the model based upon dynamical system theory revealed that the same thalamocortical mechanisms that governed perceptual awareness in threshold detection determined perceptual dominance in visual rivalry. Crucially, in addition to reproducing existing neural and behavioural findings, the model generates testable electrophysiological predictions.

**Data availability statement:** Code to reproduce the simulations reported in the paper can be downloaded from https://github.com/cjwhyte/LVPA.

**Funding:** This study was supported by the National Health and Medical Research Council (GNT1193857), Australian Research Council (DP240101295) and the Viertel/Bellberry Foundation (J.M.S). The funders had no role in study design, data collection and analysis, decision to publish, or preparation of the manuscript

**Competing interests:** The authors have declared that no competing interests exist.

## Introduction

The study of perceptual awareness – the process of gaining conscious access to perceptual content – in human participants [1–3] and animal models [4–7] have opposing but complementary limitations. Human participants can rapidly learn complex tasks that isolate and control for key psychological constructs, however the high-resolution (i.e., cell specific) recordings and precise causal manipulations (e.g., optogenetic and pharmacological) that are needed to make effective inferences about the neural basis of behaviour are exceedingly difficult and often impossible to obtain. At the same time, animal models, and transgenic mouse models in particular, allow for an astonishing degree of experimental precision in the recording and causal manipulation of neural activity. Animal models are, however, highly limited in the range and complexity of the tasks they can perform, restricting the type of psychological inferences that can be drawn. Both fields contain crucial pieces of the puzzle for understanding perceptual awareness, however the links between the two are limited at best. Effective progress, therefore, hinges on our ability to create empirically tractable tethers between the behavioural signatures of perceptual awareness studied in humans and the fine-grained neurobiological mechanisms studied in animal models [8].

Recent work in a mouse model of perception has identified a key thalamocortical circuit connecting thick-tufted layer 5 pyramidal-tract ($L5_{PT}$) neurons and matrix thalamic cells as playing a causal role in the threshold for perceptual awareness [9–12]. Specifically, based on a range of cellular recordings and causal perturbations, it has been shown that matrix-thalamus-mediated coupling of apical dendrite and somatic compartments in $L5_{PT}$ cells leads to a burst-firing state that is a reliable signature of perceptual awareness of a near-threshold tactile stimulus [12,13]. However, the simplicity of the threshold detection task and species-specific differences in neural architecture means that it is not clear whether the mechanisms of perceptual awareness characterised in the mouse model will generalise beyond the whisker detection task to the more complex paradigms typically studied in human participants.

Here, we use biophysical modelling to bridge the gap between the thalamocortical circuit identified in the mouse model of perception [10,12,13] and the behavioural signatures of perceptual awareness studied in human psychophysics. Specifically, we built a thalamocortical spiking neural network model that explains the full suite of behavioural and neural findings in the mouse model of tactile threshold detection. Given the ubiquity of the thalamocortical circuit architecture across sensory modalities, we [10,11,14,15], along with others [16,17], have proposed that reverberant bursting activity in $L5_{PT}$ – matrix thalamus loops may be a necessary component part in a domain general mechanism of perceptual awareness. A key test of this hypothesis is whether this same circuit architecture can explain psychophysical principles known to govern perceptual awareness in more complex paradigms and in other sensory modalities.

To test this hypothesis *in silico*, we leveraged the same model with identical parameters to simulate both tactile threshold detection and visual rivalry (which we use as a catch all term for binocular rivalry and related bistable perception paradigms). Visual rivalry is a complex but highly psychophysically-constrained phenomenon whereby visual perception stochastically switches between stimulus percepts that differ only in terms of their perceptual content [18–21]. Visual rivalry provides a means to dissociate the neural mechanisms of subjective perception from the correlates of physical stimulation. Crucially, variants of visual rivalry (i.e., plaid rivalry [22]) can now be studied in mouse models [7,23] as well in human and non-human primates, allowing us to test our model against existing psychophysical findings in primates and to make predictions for what should be observed in the mouse model before data has been collected – a central component in the evolving dialogue between theory and experiment. In addition, although our model is consistent with the psychophysical predictions

of previous models of visual rivalry [24–32], our approach has an unique level of neurobiological specificity that allows us to generate cellular level predictions about the neural underpinnings of perceptual awareness in a language that is applicable to the causal methods used by modern systems neuroscientists.

## Results

### A spiking corticothalamic model recreates key features of cellular physiology

The dynamical elements of the model were inspired by recent empirical observations, and consist of three classes of neurons (Fig 1A and 1B) – $L5_{PT}$ cells (blue), fast spiking interneurons (basket cells, gold), and diffuse projecting matrix thalamocortical cells (purple) – each of which are modelled using biophysically plausible spiking neurons [34–37] that were coupled through conductance-based synapses. By building the model from these circuit elements, we ensured that the emergent dynamics of the population recapitulate known signatures of cell-type-specific firing patterns, thus retaining the capacity to translate insights between computational modellers and cellular physiologists.

To model the non-linear bursting behaviour of $L5_{PT}$ cells (which has been linked to perceptual awareness [33,38,39]), we created a novel dual-compartment model with active apical dendrites that captures the essential features of the cells' physiology. Empirical recordings have shown that $L5_{PT}$ cells switch from regular spiking to bursting when they receive near-simultaneous input to both their apical (top) and basal (bottom) dendrites ([33,38]; Fig 1B). As such, our model includes a somatic compartment, described by an Izhikevich adaptive quadratic integrate and fire neuron [36,40,41] and an apical compartment, described by a non-linear model of the $Ca^{2+}$ plateau potential [37,42]. To recapitulate known physiology,

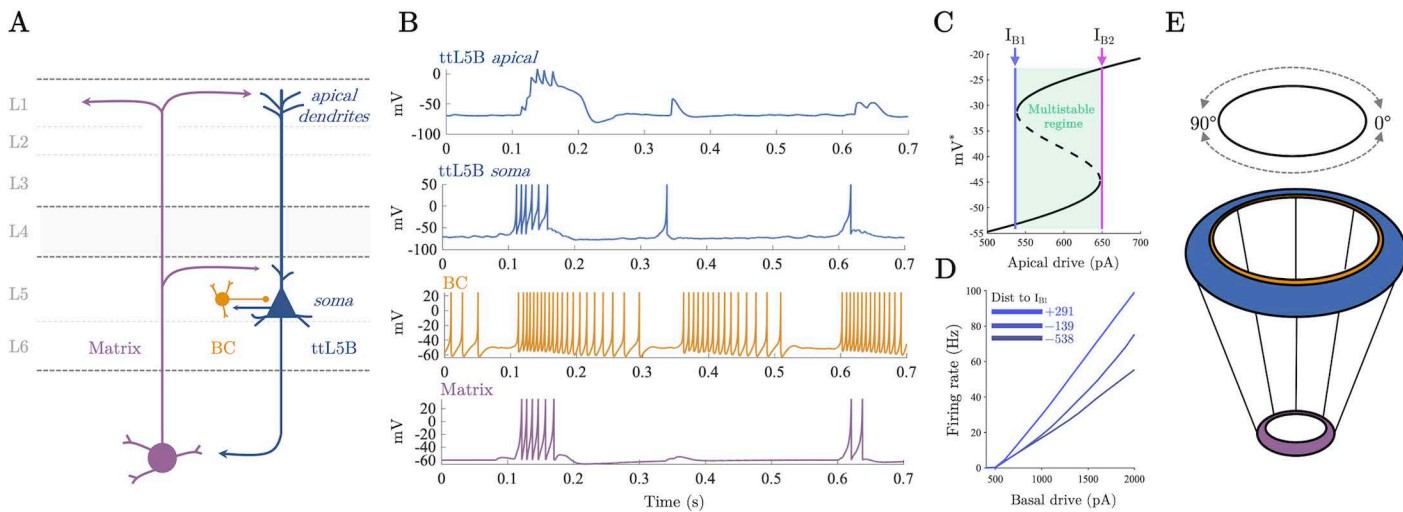

**Fig 1. Thalamocortical model of perceptual awareness. A) Idealised anatomy of the model thalamocortical loop connecting higher-order matrix thalamus and $L5_{PT}$ neurons. B) Single neuron dynamics of example neurons in the thalamocortical network for each class of cell when driven with 600 Hz of independent background drive to the somatic compartment of every neuron and 50 Hz to apical compartment of $L5_{PT}$ cells. C) Bifurcation diagram of the $L5_{PT}$ apical compartment. $I_{B1}$ denotes a saddle node bifurcation generating a stable plateau potential which coexists with the resting state of the apical compartment until the model passes through a second saddle node bifurcation at $I_{B2}$ at which point the resting state vanishes and the stable plateau potential becomes globally attracting. D) Somatic firing rate of the novel dual compartment $L5_{PT}$ model as a function of basal (i.e. somatic compartment) drive and apical drive (measured in terms of the distance to $I_{B1}$). In line with empirical data [33], apical drive increases the gain of the somatic compartment. E) Model thalamocortical ring architecture. $L5_{PT}$ cells and basket cells were placed on a cortical ring at evenly space intervals.**

we coupled the compartments such that sodium spikes in the somatic compartment back propagate to the apical compartment; in turn if a $Ca^{2+}$ plateau potential is triggered in the apical compartment the somatic compartment's behaviour (probabilistically) switches from regular spiking to bursting (implemented by switching the reset conditions of the somatic compartment). The amount of current entering the apical compartment controls this switching process. With sufficiently high current the apical compartment passes through a saddle node bifurcation ($I_{B1}$; Fig 1C) and a stable $Ca^{2+}$ plateau potential coexists with the resting state of the compartment. Further increases in current cause the cell's resting state to disappear (by passing the cell through a second saddle node bifurcation at $I_{B2}$; Fig 1C), making the plateau potential globally attracting (Fig A in S1 Appendix).

Based on the finding that communication between the soma and apical dendrites of $L5_{PT}$ cells depends upon depolarising input from the matrix thalamus to the "apical coupling zone" of $L5_{PT}$ cells [43], we made the probability of successful propagation between compartments proportional to the amplitude of thalamic conductances. In line with empirical findings and previous modelling [33,44], $Ca^{2+}$ plateau potentials in the apical compartment controlled the gain of the somatic compartment's firing rate curve by increasing the amount of time the somatic compartment spent in a bursting rather than a regular spiking parameter regime (Fig 1D).

In line with previous spiking neural network models of early sensory cortex [25,26,45,46], we embedded the cortical neurons in a one-dimensional ring architecture (90 pairs of $L5_{PT}$ excitatory and fast-spiking inhibitory interneurons; Fig 1E). Each point on the ring represents an orientation preference, with one full rotation around the ring corresponding to a 180° visual rotation – this provides each neuron with a 2° difference in orientation preference, relative to its neighbours. The cortical ring was coupled to a thalamic ring with a 9:1 ratio (to approximately reflect the cortico-thalamic ratio in mammals), which then projected back up to the apical dendrites of the same 9 cortical neurons, representing the diffuse projections of higher-order thalamus onto the apical dendrites of $L5_{PT}$ neurons in layer 1 [47,48]. Cortical coupling was modelled with a spatial decay, with long range inhibitory coupling and comparatively local excitatory coupling (i.e., centre-surround 'Mexican-hat' connectivity).

When driven solely by baseline input, the model emitted irregular spikes interspersed with sparse spatially localised bursts mediated by depolarising input from the thalamus which allowed $L5_{PT}$ somatic spikes to back-propagate initiating $Ca^{2+}$ plateau potentials in the apical dendrites which in turn initiated transient burst spiking in the somatic compartment (Fig 1C). Cortical spiking activity was highly irregular (mean inter-spike interval coefficient of variation $\frac{standard\,deviation}{mean}$ = 2.4) characteristic of a waking state [49]. For more details on the model architecture and analysis of the dynamics see **materials and methods**.

## The thalamocortical model reproduces empirical signatures of threshold detection

We first set out to reproduce the results of the whisker-based tactile detection paradigm employed by Takahashi and colleagues [12,13], who trained mice to report a mechanical deflection of a whisker over a range of deflection intensities (Fig 2A) while recording $L5_{PT}$ activity in barrel cortex from the apical dendrites via fast scanning two-photon $Ca^{2+}$ imaging, and somatic activity via juxtacellular electrodes. They found that bursting activity in the soma of $L5_{PT}$ cells, generated by $Ca^{2+}$ plateau potentials in the apical dendrites, distinguished hits and false alarms from misses and correct rejections. Importantly, they were able to establish causality through a series of perturbation experiments (Fig 2B). Optogenetic excitation of the apical dendrites reduced the animal's threshold for awareness increasing both hits and

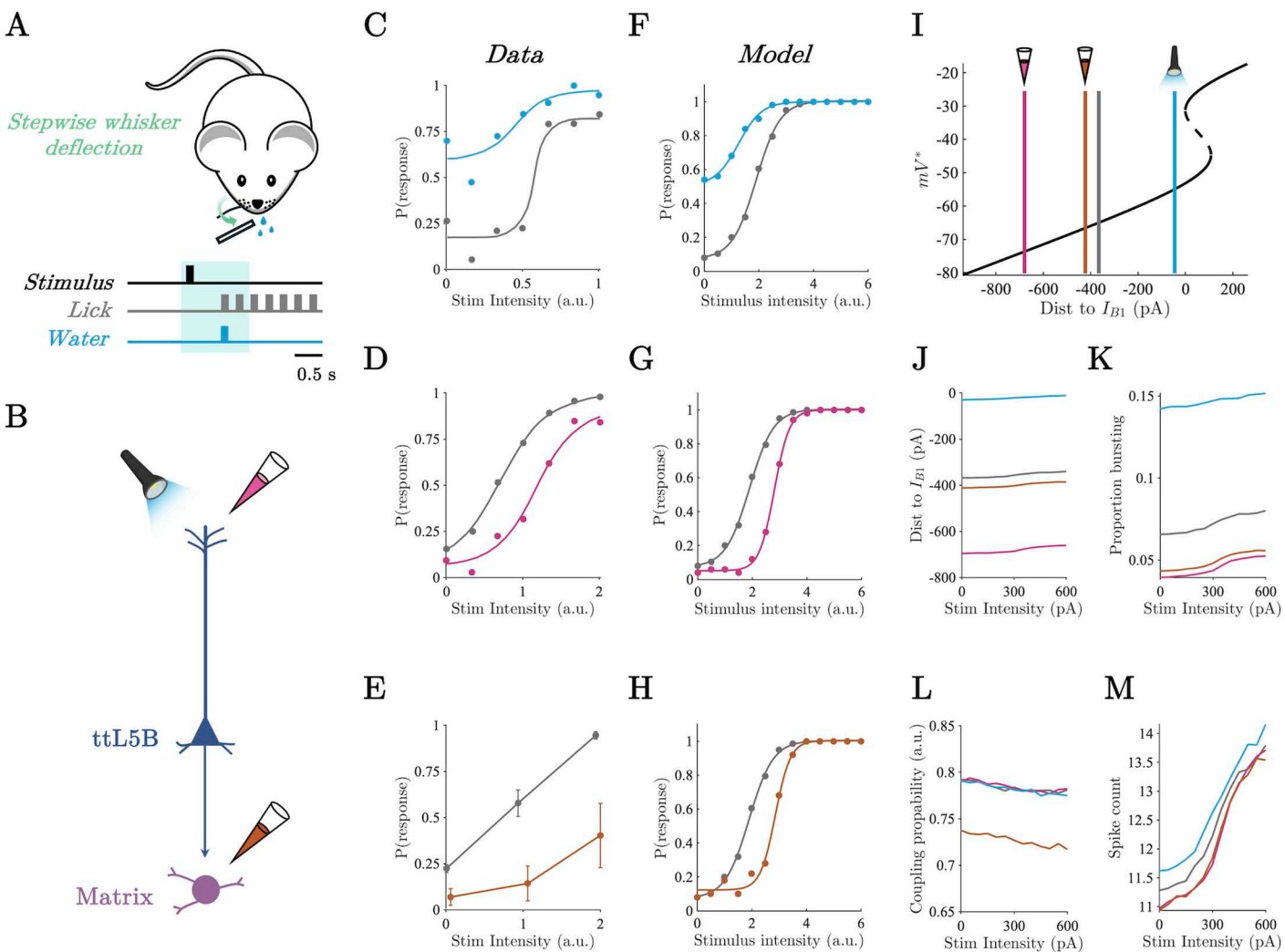

**Fig 2. Apical compartment distance to bifurcation and thalamic-gating explains shifts in perceptual threshold. A)** Whisker deflection paradigm of Takahashi and colleagues [12]. **B)** Representation of causal perturbations to L5$_{PT}$ thalamocortical circuit including optogenetic excitation of the apical dendrites (light blue), pharmacological inhibition of the apical dendrites (pink), and pharmacological inhibition of POm (orange). **C-D)** Psychometric function of animals performing a whisker deflection task for the control condition (grey), optogenetic excitation of apical dendrites (blue), and pharmacological inhibition of apical dendrites (pink). Modified from Takahashi and colleagues [12,13]. **E)** Response probability as a function of whisker deflection intensity for control (grey) and pharmacological inhibition of POm (orange). Modified from Takahashi and colleagues [12]. **F-H)** Model response probability as a function of stimulus intensity across simulated causal perturbations. Colours same as above. **I)** Apical compartment bifurcation diagram showing the average distance to B$_1$ in post stimulus period averaged across stimulus intensities. **J)** Average distance to B$_1$ across stimulus intensities and simulated causal perturbations in the post stimulus period. **K)** Proportion of population in bursting regime across stimulus intensities and simulated causal perturbations in the post stimulus period. **L)** Average inter-compartment coupling probability across stimulus intensities and simulated causal perturbations in the post stimulus period. **M)** Average spike count across stimulus intensities and simulated causal perturbations in the post stimulus period.

false-alarms (Fig 2C). In turn, pharmocological inhibition of the apical dendrites and POm (a matrix-rich higher-order thalamic nucleus with closed loop connections to barrel cortex [47]) increased the animals perceptual threshold (Fig 2D and 2E).

To model the perceptual discrimination process underlying threshold detection –discriminating the presence of a weak stimulus against a noisy background – all neurons received a constant background drive consisting of independent Poisson spike trains while an arbitrary cortical neuron was pulsed by a current of constant width and variable amplitude that we weighted by a spatial Gaussian to mimic the selectivity of neurons in early sensory cortex.

We operationalised perceptual awareness in the threshold detection simulations in terms of what a downstream ideal observer could readout from the population by computing whether trial-by-trial spike counts in the 1000 ms post stimulus window exceeded an optimal criterion (i.e. the criterion that best minimised misses and false alarms across stimulus intensities). We counted the model as having made a response whenever the spike count exceeded the optimal criterion. Psychometric functions were then fit to the model' responses (for details see **materials and methods**). Qualitatively identical results were obtained using neurometric functions which summed over all criterion values (see Fig B in S1 Appendix [50]).

The model responses to simulated whisker deflections recapitulated the empirically observed sigmoidal relationship between the intensity of the simulation and the model's response probability (Fig 2F-H). In addition, perturbations to the model designed to replicate optogenetic excitation and pharmacological inhibition (Fig 2A) qualitatively reproduced the empirically observed shifts in response probability across perturbation types (Fig 2C-H). For each type of perturbation we ran the simulation over a range of perturbation magnitudes to ensure the reliability of the effect see Fig B in S1 Appendix. For brevity, we only show the results for 300 pA perturbations in the main text. In addition, we note that the saturating detection probability values in the model are due to an absense of behavioural stochasticity resulting from extranious factors such as decision noise which are inherent to empirical data.

A key benefit of biophysical modelling is the capacity to mechanistically probe the model and determine how the empirical observations may have emerged from the underlying circuit dynamics. To this end, we used tools from dynamical systems theory to interrogate the cell-type-specific dynamics underlying the behaviour of the model including the distance to bifurcation ($I_{B1}$) in the apical compartment, the parameter regime of the somatic compartment (i.e. regular spiking or bursting), and the inter-compartment coupling probability. Excitation of the apical dendrites (blue) reduced the average distance to bifurcation (here defined as the distance to $I_{B1}$) in the apical compartment across the network in the 1000 ms period post stimulus onset (Fig 2I and 2J). This increased the proportion of time each somatic compartment spent in the bursting regime (Fig 2K), which in turn increased the average spike count of $L5_{PT}$ cells across stimulus intensities resulting in reduction of the model's perceptual threshold (Fig 2F and 2M). Conversely, inhibition of the apical dendrites (pink) and the thalamus (orange) both resulted in an increase in the perceptual threshold (Fig 2G and 2H).

Importantly, however, the mechanisms underlying the increase in the perceptual threshold differed across apical dendrite and thalamic inhibitory perturbations. Inhibition of the apical dendrites increased the average distance to bifurcation at $B_1$ in the apical compartment across the network (Fig 2I and 2J). In contrast, inhibition of the thalamus resulted in a comparatively minor reduction in the distance to bifurcation in the apical dendrites (Fig 2I and 2J) but reduced the thalamus mediated inter-compartment coupling (Fig 2L) thereby reducing the probability that a back propagating action potential could reach the apical compartment, and the probability with which a plateau potential could switch the regime of the somatic compartment from regular spiking to bursting. Together this resulted in a similar reduction in the proportion of cells in the bursting regime for both thalamic and apical dendrite inhibition, and likewise, a similar reduction in the average stimulus evoked spike count explaining the comparable increase in perceptual thresholds (Fig 2M).

## Thalamocortical spiking model generalises to visual rivalry

We next sought to generalise our thalamocortical model of perceptual awareness to visual rivalry formalising and interrogating the hypothesis that the role played by pulvinar – $L5_{PT}$ loops in visual cortex is analogous to the role played by POm – $L5_{PT}$ loops in barrel cortex. To simulate visual rivalry we drove the model with input representing orthogonal gratings

presented to each eye, typical of standard binocular rivalry experiments [21,51], targeting the soma (i.e., basal dendrites) of L5$_{PT}$ cells on opposite sides of the ring with orthogonal orientation preferences (Fig 3A).

Due to the fact that inhibitory connectivity is broader than excitatory connectivity, delivering external drive to opposite sides of the ring shifts the model into a winner-take-all regime with burst-dependent persistent states on either side of the ring competing to inhibit one another. Importantly, through the accumulation of slow a hyperpolarising adaptation current in the somatic compartment of each L5$_{PT}$ cell (representing slow Ca$^{2+}$ mediated K$^+$ currents [52,53]), burst-dependent persistent states are only transiently stable leading to stochastic

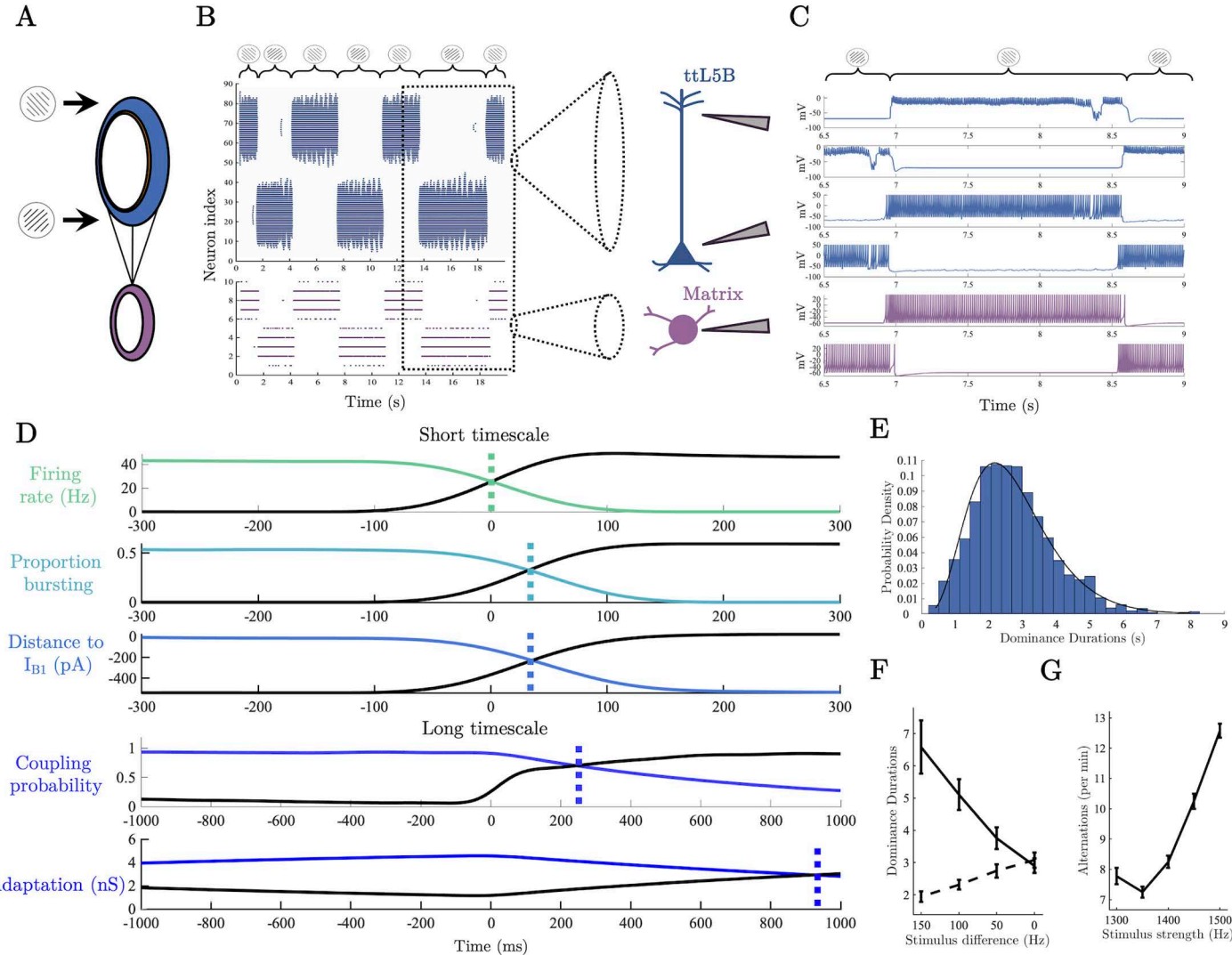

**Fig 3. A thalamocortical cascade underlies perceptual switches. A)** Thalamocortical ring architecture driven by input representing orthogonal gratings. **B)** Raster plots of L5$_{PT}$ soma (blue) and matrix thalamus population (purple) during rivalry. **C)** Example single neuron spiking activity around a perceptual switch – dotted lines denote the crossing in the population averaged time series. **D)** Population averaged neuronal variables centred on a perceptual switch. **E)** Histogram of dominance durations, black line shows the fit of a Gamma distribution with parameters estimated via MLE ($\alpha$ = 4.85, $\theta$ = 0.56). **F)** Simulation confirming Levelt's second proposition. Dashed line shows the dominance duration of the population receiving the decreasing external drive, solid line shows dominance duration of population receiving a fixed drive. **G)** Simulation of Levelt's fourth proposition. Notably, for low stimulus strengths our model, along with a number of meanfield models of visual rivalry [28], predicts a deviation from Levelt's fourth proposition.

switches between persistent states characteristic of binocular rivalry (Fig 3B and 3C). We operationalised perceptual dominance (i.e., awareness of a percept at the exclusion of the other) in terms of the difference in average firing rate between the persistent 'bumps' of $L5_{PT}$ cell activity on opposite sides of the ring. In line with common practice (e.g [54]), a population was counted as dominant when it had a firing rate 5 Hz greater than its competitor and lasted for longer than 250 ms (results were robust across a large range of threshold values). The initial competition for perceptual dominance was determined by which population of $L5_{PT}$ cells first established a recurrent loop with matrix-thalamus. This interaction allowed the recurrently connected population of cells to enter a bursting regime, at which point the population had sufficient activity to maintain a persistent state and inhibit the competing population into silence. In previous work we have argued that this emergent property of corticothalamic loops is a good candidate for a cellular-level correlate of perceptual awareness [10,15].

Crucially, our model provides a cellular-level explanation of spontaneous rivalry-induced perceptual switches in terms of these same mechanisms. As the reliable initiation of $Ca^{2+}$ plateau potentials in the model depends upon back-propagating action potentials, perceptual switches are always preceded by regular spikes in the suppressed population. Preceding each switch, the accumulation of adaptation reduces the firing rate of the dominant population to a sufficient level that the suppressed population can escape from inhibition and emit a brief series of regular spikes that transition into bursts as thalamus is recruited and $Ca^{2+}$ plateau potentials are initiated (Fig 3C), eventually gaining sufficient excitatory activity to inhibit the previously dominant population into silence via recurrent interactions with the basket cell population. At the population level (Fig 3D), this cascade of neuronal events is characterised by an initial ramping in the somatic firing rate of the suppressed population, followed by the inter-compartment coupling probability, the proportion of the population bursting, and finally by the average distance to bifurcation in the apical compartment. Once dominant, approximately half the population is in a bursting regime at each point in time (0.5424 mean $\pm$ 0.058 standard deviation), and the average distance to bifurcation in the apical compartment fluctuates around zero (-11.0185 $\pm$ 48.067 pA), with approximately a third (0.3669 $\pm$ 0.097) of the population located above the critical boundary ($I_{B1}$) at each point in time. In addition, the average inter-compartment coupling probability is (0.8845 $\pm$ 0.102) allowing reliable communication between compartments. Once silenced, adaptation in the previously dominant population decays back to baseline levels and accumulates in the previously suppressed, now dominant, population. In this way, the competitive organisation of the cortico-cortical and corticothalamic loop provides a natural "flip-flop" switch that stochastically alternates between dominant percepts.

As cortical pyramidal cells are known to display considerable differences in spiking behaviour across species [55] we swept the parameters of our model $L5_{PT}$ cell responsible for the bursting behaviour, and the inter-compartment coupling probability, to ensure that visual rivalry (quantified by average dominance duration) is stable across a wide range of parameters. In favour of the robustness of the model, as long as there was reliable inter-compartment coupling, which we assume is the case in the waking state, dominance durations in an empirically plausible range (i.e. with a period on the order of seconds) were present across a wide range of parameter values (Fig C in S1 Appendix).

In close agreement with psychophysical data, switches between dominance and suppression had a right-skewed distribution of dominance durations ([51,56,57]; Fig 3E). Across stimulus drive conditions (1300 – 1500 Hz), comparison of negative log likelihoods showed that the distribution of dominance durations was best fit by a Gamma distribution ($\ell = 2.81 \times 10^{-3}$), compared to lognormal ($\ell = 2.88 \times 10^{-3}$) or normal distributions ($\ell = 2.88 \times 10^{-3}$). Because of the large size of the network and the discontinuity

in the somatic compartment we could not use analytic methods to interrogate the structure of the dynamical system underlying the stochastic oscillations. Instead we used a heuristic line of argument (see [58], Ch.8) combined with simulations in the absence of noise [59] to confirm that the dynamical regime underlying these stochastic oscillations likely consists of noisy excursions around a stable closed orbit (i.e. a stable limit cycle; Fig D in S1 Appendix).

## Thalamocortical spiking model conforms to Levelt's propositions

To further test the psychophysical validity of our neurobiologically detailed model, we simulated the experimental conditions described by Levelt's modified propositions [56] – a set of four statements that compactly summarise the relationship between stimulus strength (e.g., luminance contrast, colour contrast, stimulus motion) and the average dominance duration of each stimulus percept. Here, we focus on the modified second and fourth propositions, as they constitute the "core laws" of rivalry and incorporate recent psychophysical findings (propositions one and three are consequences of proposition two [56]).

Levelt's modified second proposition states that increasing the difference in stimulus strength between the two eyes will principally increase the average dominance duration of the percept associated with the stronger stimulus [56,60]. To simulate Levelt's second proposition, we decreased the spike rate entering one side of the ring from 1400 to 1250 Hz in steps of 50 Hz across simulations. In line with predictions (Fig 3F), the average dominance duration of the percept corresponding to the stronger stimulus showed a steep increase from ~3 s with matched input, to ~ 6.5 s with maximally different input while the average dominance duration on the side of the weakened stimulus decreased comparatively gradually to ~2 s with maximally different inputs.

According to Levelt's modified fourth proposition, increasing the strength of the stimulus delivered to both eyes will increase the average perceptual reversal rate (i.e., decrease dominance durations [56]), a finding that has been replicated across a wide array of experimental settings [61–64]. To simulate Levelt's fourth proposition, we ran a series of simulations in which we increased the spike rate of the external drive in steps from 1300 to 1500 Hz in steps of 50 Hz across simulations. Again in line with predictions (Fig 3G), the perceptual alternation rate increased with input strength, starting at ~7.75 alternations per minute at the second weakest stimulus strength (1350 Hz) and increasing to ~15 alternations per minute for the strongest stimulus (1500 Hz). Interestingly, along with a number of meanfield models of rivalry [28], our model predicts a deviation from Levelt's fourth proposition for very low stimulus values with an uptick in alternation rate occurring at the lowest external dive value (1300 Hz). Encouragingly, there is some initial evidence that deviations from Levelt's fourth law may be present in human psychophysical data [56].

To help ensure that the simulation results were not biased by finite size effects or other simplifying assumptions such as the all-to-all connectivity of the cortical ring, or the 50/50 excitatory/inhibitory neuron ratio, we show in Fig E in S1 Appendix that a scaled-up version model consisting of 2000 cortical neurons with sparse connectivity, and an 80/20 excitatory/inhibitory neuron ratio (i.e. consistent with Dale's law), also produces a Gamma distribution of dominance durations, and is consistent with Levelt's second and fourth propositions.

We thus confirmed that our neurobiologically detailed model of the matrix thalamus - $L5_{PT}$ loop is capable of reproducing Levelt's propositions, which together with the right-skewed distribution of dominance durations, show the consistency of our model with the psychophysical "laws" known to govern visual rivalry.

## Generating testable predictions through in silico electrophysiology

Binocular rivalry is thought to depend in part on the substantial degree of binocular overlap in humans (~120°), however the lateral position of the eyes in mice leaves only ~40° of binocular overlap [65]. For this reason, there are no current mouse models of binocular rivalry, however there are monocular variants of visual rivalry, namely plaid perception, that can be studied the mouse model [7,66]. Crucially, plaid perception, like binocular rivalry, conforms to Levelt's laws [56,67] and also has a right skewed distribution of dominance durations that is well fit by a Gamma distribution [66]. We hypothesise, therefore, that the principles underlying the simulation of binocular rivalry in our model will also describe other forms of visual rivalry (such as plaid perception), offering a plausible means to test cellular level predictions derived through simulation. To this end, we next ran a series of perturbation experiments, with the aim of interrogating the novel burst-dependent mechanism of perceptual dominance by mimicking the optogenetic and pharmacological experiments carried out in threshold-detection studies [12,13], in the context visual rivalry. As perceptual dominance depends on the formation and maintenance of a burst-dependent persistent state, we hypothesised that artificially exciting the apical dendrites would result in an increase in the average dominance duration of the excited population, and artificially inhibiting the apical dendrites and thalamus would result in a decrease in the dominance durations for the inhibited populations.

Due to the fact that the distance to bifurcation of the dominant population fluctuates around zero, we predicted that exciting the apical dendrites would increase the proportion of the population above the bifurcation at $B_1$, thereby reducing the probability with which a fluctuation in somatic drive would lead to a sizable enough drop in the proportion of the population below $B_1$ to release the competing population from inhibition. This should, therefore, result in an increase in the frequency of long dominance duration events, thereby increasing the mean and the spread of the distribution. Equivalently, we predicted that inhibiting the apical dendrites would reduce the proportion of the population above $B_1$, making it more likely that transient fluctuations in somatic drive would allow the competing population to escape from inhibition, reducing the occurrence of long dominance duration events, thereby reducing both the mean and the spread of the distribution of dominance durations. Finally, based on the results of the threshold detection simulations we predicted that thalamic inhibition would have an analogous effect on dominance durations to apical dendrite inhibition but would be mediated by a reduction in the coupling probability. To test these hypotheses, we conducted two *in silico* experiments analogous to the conditions described by Levelt's modified propositions but instead of manipulating the external drive entering the somatic compartment of $L5_{PT}$ cells we manipulated the amplitude of simulated causal perturbations to the $L5_{PT}$ apical compartment and thalamus (Figs 4A and 5A).

In the first set of experiments, we simulated optogenetic excitation and pharmacological inhibition of one of the two competing populations by adding a constant current ($\pm$ 200, 400 pA) to all of the target variables (i.e. apical compartment or thalamic neurons) on one side of the ring (Fig 4A). In line with predictions, we found that the average dominance duration of the excited population (Fig 4B) increased, the distance to bifurcation decreased (Fig 4E and 4F), the proportion of the population above the critical point at $B_1$ increased (Fig 4G), and the proportion of the population in the bursting regime increased (Fig 4H). The dominance durations and neuronal dynamics of the unexcited population remained relatively unchanged.

Similarly, inhibition of both the apical dendrites and thalamus reduced the average dominance duration of the inhibited population while the uninhibited population was again relatively unchanged (Fig 4C and 4D). As in the threshold detection simulations, inhibition of the apical dendrites led to a large increase in the distance to $B_1$ compared to thalamic inhibition

(Fig 4E and 4F) which primarily affected the inter-compartment coupling probability (Fig FA in S1 Appendix). Both apical dendrite and thalamic inhibition led to almost identical reductions in the proportion of the population above the critical point at $B_1$ (Fig 4G), and the proportion of the population in the bursting regime (Fig 4H). The uninhibited population again remained relatively constant across all of the neuronal measures (the small drop in dominance durations of the uninhibited population for the 400 pA inhibitory perturbations is due to the adaptation variable having less time to recover). As predicted the spread of the distribution of dominance durations increased with the amplitude of excitatory perturbation and decreased under inhibition (Fig FB in S1 Appendix).

In the second set of experiments, we simulated optogenetic excitation and pharmacological inhibition of both competing neuronal populations simultaneously by adding a constant current ($\pm$ 200, 400 pA) to all of the target variables on the ring (Fig 5A). Again, in line with predictions, the speed of rivalry (i.e., the number of perceptual alternations per minute of simulation time) decreased as a function of apical dendrite excitation (Fig 5B). Excitation also decreased the average distance to $B_1$ (Fig 5E and 5F), increased the proportion of the population above the critical point at $B_1$ (Fig 5G), and increased the proportion of the population in the bursting regime (Fig 5H).

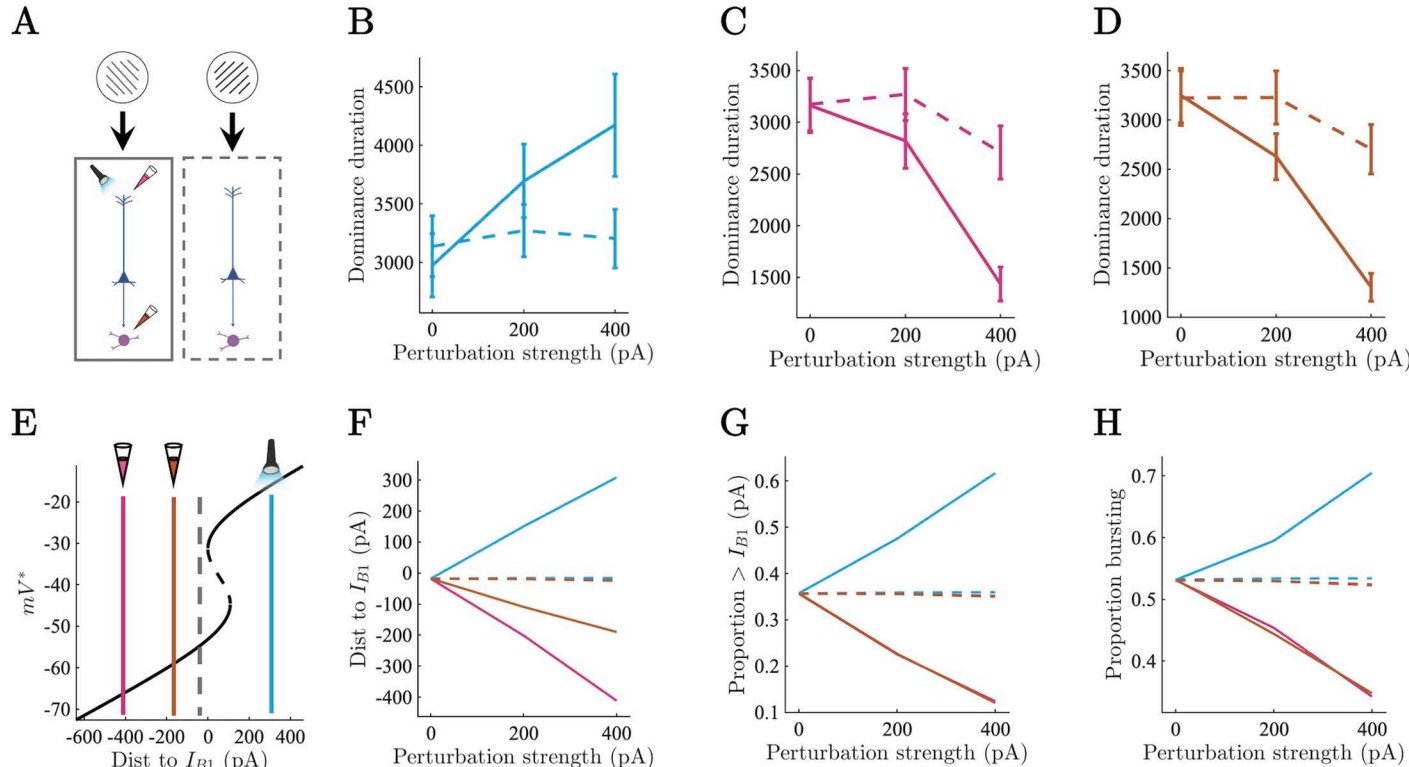

**Fig 4. *In silico* electrophysiology reveals degenerate mechanisms of perceptual dominance (asymmetric perturbations). A)** Causal perturbations to one half of the thalamocortical circuit underlying visual rivalry consisting of optogenetic excitation of the apical compartment (blue), pharmacological inhibition of the apical compartment (pink), and pharmacological inhibition of the thalamus (orange). **B-D)** Average dominance duration of perturbed (solid), and unperturbed (dashed) populations. Error bars show SEM. **E)** Average distance to bifurcation point at $B_1$ shown on bifurcation diagram for perturbed (solid) and unperturbed (dashed) populations during periods of perceptual dominance with 400 pA perturbation strength. **F)** Average distance to bifurcation point at $B_1$ for perturbed (solid) and unperturbed (dashed) populations during periods of perceptual dominance. **G)** Proportion of population above bifurcation point at $B_1$ during periods of perceptual dominance. **H)** Proportion of population in bursting regime during periods of perceptual dominance.

In contrast, inhibition of the apical dendrites and thalamus both increased the speed of rivalry (Fig 5C and 5D). Inhibition of the apical dendrites led to a large increase in the distance to the critical point at $B_1$ compared to thalamic inhibition (Fig 5E and 5F), but both apical dendrite inhibition and thalamic inhibition reduced the proportion of the population above the critical point at $B_1$ (Fig 5G), and the proportion of the population in a bursting regime (Fig 5H) through reductions in the thalamus mediated inter-compartment coupling probability (Fig FC in S1 Appendix). As with the asymmetric perturbation simulations, inhibition exerted a much larger effect on the speed of rivalry than excitation. Finally, again in line with our predictions, the spread of the distribution of dominance durations increased under excitatory perturbation of the apical dendrites and decreased under inhibition of the apical dendrites and thalamus (Fig FD in S1 Appendix).

Together, these *in silico* electrophysiological experiments provide important testable (and explainable) hypotheses for future experiments that although not testable in any existing data sets are well within the purview of modern systems neuroscience providing an opportunity to conduct precise theory-driven tests of the model.

## Discussion

The study of perceptual awareness in human participants and animal models has so far proceeded largely in parallel – the former exploring the largescale neural dynamics and behavioural signatures of perceptual awareness across a rich array of experimental settings,

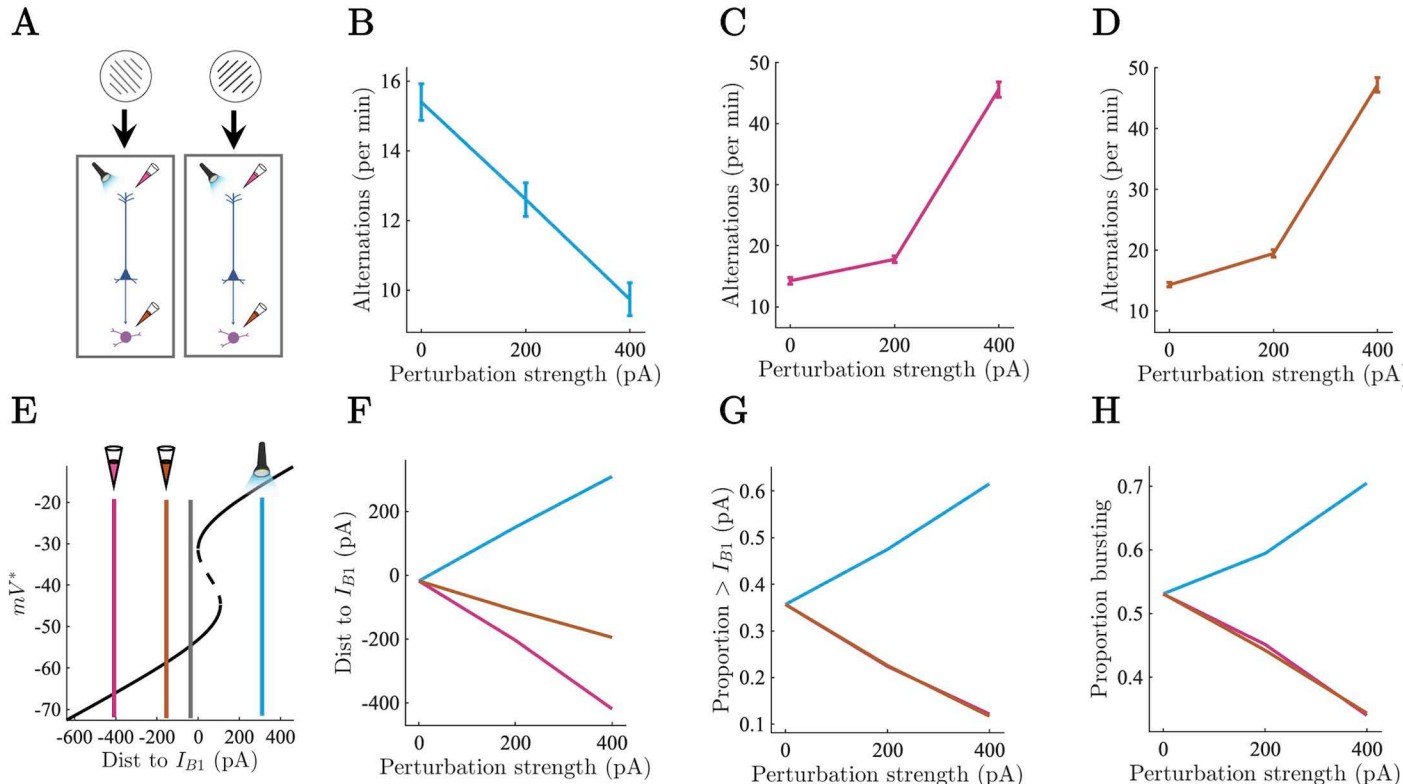

**Fig 5. *In silico* electrophysiology reveals degenerate mechanisms of perceptual dominance (symmetric perturbations). A)** Causal perturbations to full thalamocortical circuit underlying visual rivalry. Colours same as above. **B-D)** Perceptual alternations per minute of simulation time across perturbation types. Error bars show SEM. **E)** Average distance to bifurcation point at $B_1$ shown on bifurcation diagram for perturbed (blue, orange, pink) and unperturbed (grey) simulations during periods of perceptual dominance at 400 pA. **F)** Average distance to bifurcation point at $B_1$ for full network perturbations during periods of perceptual dominance. **G)** Average proportion of population above bifurcation point at $B_1$ for full network perturbations during periods of perceptual dominance. **H)** Average proportion of population in bursting regime for full network perturbations during periods of perceptual dominance.

and the latter characterising the cellular circuitry of perception in exquisite detail, and with precise causal control, but with only limited links to higher level perceptual phenomena [8]. Leveraging a neurobiologically detailed model of the matrix thalamus – $L5_{PT}$ loop, we have shown that a potential circuit-level mechanism of tactile perceptual awareness discovered in a mouse model of tactile awareness [9,12,13] generalises to visual rivalry, thus providing a roadmap for the linking circuit level mechanisms studied in animal models to the behavioural signatures of perceptual awareness studied in human participants.

The balance of neurobiological detail and interpretability offered by our model allowed us to reproduce the threshold-detection results of Takahashi and colleagues [12,13] and interrogate the mechanisms underlying the experiments in a manner that would be impossible *in vivo*. In particular, examination of the model's dynamics under simulated causal perturbations to the circuit revealed a degenerate dynamical mechanism for controlling the threshold for perceptual awareness. Excitation of the apical compartment reduced the distance to bifurcation in the apical compartment, thus increasing the probability that each cell could generate a $Ca^{2+}$ plateau potential switching the soma into a bursting regime. This resulted in an increase in the baseline and stimulus-evoked spike count, and correspondingly, led to a reduction in the model's perceptual threshold. Inhibition of the apical compartment and thalamus resulted in comparable downward shifts in the baseline and stimulus-evoked spike count, leading to increases in the model's perceptual threshold. Importantly, however, the neural mechanisms underlying the increases in perceptual threshold were distinct: inhibiting the apical compartment increased the distance to bifurcation, thus reducing the probability with which each cell would generate a $Ca^{2+}$ plateau potential, whereas inhibiting the thalamus reduced the inter-compartment coupling. Both mechanisms, however, led to comparable reductions in the proportion of cells in the bursting regime explaining the comparable increase in perceptual thresholds, suggesting that it is the emergent action of the corticothalamic circuit as a whole, rather than single cells within the circuit, that are responsible for perceptual awareness.

The degenerate mechanisms underlying the threshold for perceptual awareness combined with the operational definition of perceptual awareness in the threshold detection task (in terms of psychometric functions) points to a conceptually important point about the role of bursts in the model, and potentially, the empirical data itself. Specifically, controlling the ease with which a cell can burst through optogenetic and pharmacological perturbation is simply a means for controlling how easily a stimulus can evoke reverberant activity in corticocortical and thalamocortical loops which, in the simple case of threshold detection, constrains the extent to which stimulus evoked activity can stand out against a background of noise driven fluctuations.

We next showed that the same thalamus-gated burst-dependent mechanism underlying perceptual awareness in simulations of the tactile threshold detection task also determines perceptual dominance in simulations of visual rivalry. Specifically, perceptual dominance is initiated by a succession of regular spikes and maintained through the formation of a transiently stable burst-dependent persistent state characterised by reliable coupling between apical and somatic compartments. This allows the apical compartment to generate temporally extended plateau potentials in a large subset of the dominant population reliably switching the $L5_{PT}$ soma from a regular spiking to a bursting regime. Perceptual dominance is then maintained until the slow hyperpolarising adaptation current accumulates to a sufficiently high level that the dominant population is no longer able to maintain inhibit the competing and a perceptual switch ensues.

Importantly, the model conforms to Levelt's modified propositions. Originally proposed in 1965 [68], "Levelt's laws" have proven to be remarkably robust needing only minor modification and contextualisation [56] and have, therefore, served as a benchmark for computational

models of visual rivalry (e.g. [24,26,28,31]). Together with the right-skewed (Gamma) distribution of dominance durations the consistency of our model with Levelt's propositions provides an *in silico* conformation of the hypothesis that pulvinar – $L5_{PT}$ loops in visual cortex may play an analogous role to POm – $L5_{PT}$ loops in barrel cortex. This is a minimal but necessary first step in testing the hypothesis that reverberant activity in matrix thalamus – $L5_{PT}$ loops is a necessary component part in a domain general mechanism of perceptual awareness.

Having validated our model against psychophysical benchmarks, we next sought to interrogate the novel thalamus-gated burst-dependent mechanism of perceptual dominance by emulating the optogenetic and pharmacological experiments carried out by Takahashi and colleagues [12,13] in the context of visual rivalry. Under conditions of visual rivalry, the simulated causal perturbations are similar to the conditions described by Levelt's propositions, but instead of manipulating the strength of the external stimulus we manipulated the strength apical compartment excitation/inhibition, or thalamic inhibition, highlighting the unique contribution of these neurobiological components to visual rivalry. Across asymmetric and symmetric perturbations excitation of the apical compartment slowed perceptual alternations (i.e., increased dominance durations) by increasing the proportion of the population able to sustain temporally extended $Ca^{2+}$ plateau potentials and remain in a transiently-stable bursting regime, whereas inhibition of both the apical dendrites and thalamus had the opposite effect. Although technically difficult, these simulated experimental manipulations are well within the purview of modern experimental techniques and therefore represent a means of causally testing the predictions of our model. Importantly, the simulation of these experimental perturbations would not be possible in any existing models of rivalry, even those at the spiking level (e.g [26,30,45]), as they focus on the minimal conditions for rivalry in point-neuron models of cortical interaction. The inclusion of a dual compartment model of $L5_{PT}$ cells, and an explicit thalamic population was, therefore, required in order to make contact with the results of Takahashi and colleagues [12,13]. Indeed, although (thalamus-gated) plateau potential induced bursting is necessary for rivalry in our model, if we were to coarse grain the model, the dynamics would be well described by a mean field model tracking the mean firing rate of each population. Thalamic control of $L5_{PT}$ bursting is therefore not strictly speaking necessary for modelling the mesoscale dynamics of binocular rivalry, but it is necessary if one wants to capture the neurobiological processes that underpin the mesoscale dynamics and how cellular level interventions such as optogenetic excitation or pharmacological inhibition would impact rivalry dynamics as we have done here.

In addition to the predicted effect of causal perturbations on visual rivalry, our model generates a number of more straightforward correlational predictions. Specifically, matrix-rich higher-order thalamic nuclei with recurrent connections to sensory cortex, such as the pulvinar, should be selective for perceptual awareness rather than physical stimulation, a prediction supported by both human neuroimaging [69,70] and non-human primate electrophysiology [71]. Similarly, synchronous bursting activity in deep layers of cortex, specifically layer 5b which contain the soma of $ttL5_{PT}$ cells, should likewise be selective for perceptual awareness rather than physical stimulation a prediction that, with the advent of primate Neuropixels [72], is also readily testable. Finally, in the context of visual rivalry, perceptual dominance should be characterised by elongated $Ca^{2+}$ plateau potentials in the apical dendrites of $L5_{PT}$ cells (located in L1) in cells selective for the dominant percept, a prediction testable in mouse models of visual rivalry (e.g [66].).

We anticipate that the cellular conditions for awareness explored in this paper are likely to have consequences for the largescale correlates of awareness. Indeed, we venture that at the level of large scale brain networks diffuse matrix-thalamus gated bursting may play a key role in the formation of a quasi-critical regime [73,74] allowing single nodes in a network to

transiently escape from a tight E/I balanced state. This permits stimulus information to rapidly propagate across the cortical sheet while also maintaining stability at the level of the whole network [73] effectively modulating the gain of interareal connectivity in line with previous computational models of pulvinar-cortical interactions in cognitive tasks [75]. Indeed, efforts to test the largescale consequences of the cellular level mechanisms interrogated in this paper are already very much underway. Biophysical modelling of source-localised MEG data showed that auditory awareness evoked activity was best fit by increased input to superficial layers of the cortical column consistent with the projections of matrix-type higher-order thalamus [76].

As has been noted elsewhere [10,77], the circuit level conditions for awareness explored here fit well with many of the major neuronal theories of consciousness. The diffuse projections of the matrix-type thalamus may be a circuit level mechanism underlying the non-linear and widespread "ignition" response proposed by global neuronal workspace theory to underlie the transition from unconscious to conscious processing [78–81]. The improvement in signal-to-noise ratio associated with bursting aligns with signal detection theoretic versions of higher-order theory [82], and the recurrent nature of matrix thalamus – $L5_{PT}$ loops could be considered a thalamocortical extension of the currently corticocentric recurrent processing theory [83]. In addition, in previous work we have shown that diffuse matrix-type control of bursting in a sheet of $L5_{PT}$ cells maximises an approximate measure of integrated information [40], in line with integrated information theory [84]. We speculate that exploring the interaction between the cellular conditions for awareness interrogated in this paper, and the topology of largescale brain networks, may be of crucial importance in resolving the ongoing debate between the theories described above regarding the macroscale network conditions necessary for awareness.

To strike the right balance between neurobiological detail and interpretability, we made a number of simplifying assumptions that place some limitations on our model. Most notably, we did not include L2/3 pyramidal neurons – which are arguably the primary source of long distance horizontal connections in the cortex [85] and arguably cross column inhibition [69] – nor a core thalamic population which forms a targeted recurrent loops with L4 and L6 of cortex [86] preventing us from performing systematic perturbation experiments on our model highlighting the precise function of $L5_{PT}$ cells and higher-order matrix thalamus in a more realistic cortical microcircuit. We also did not include time delays between our corticocortical or thalamocortical connections preventing our model from providing a realistic model (e.g. [87]) of time-frequency components of common electrophysiological measures such as local field potentials. Finally, our model has only a single hierarchical level preventing us from making contact with evidence showing a potential prefrontal contribution to perceptual switches [88,89].

Our model is, of course, only a first step towards a formal characterisation of the minimal neurobiological mechanisms underlying perceptual awareness. Extending the model, and modelling strategy more generally, to new paradigms such as backward masking [5], will be of paramount importance in the progression of the field as mouse models and the tools of systems neuroscience are brought into contact with the sophisticated psychophysical paradigms used to study the behavioural signatures of awareness in humans.

## Materials and Methods

### Thalamocortical spiking neural network

The neuronal backbone of the model consists of a (novel) dual compartment model of $L5_{PT}$ neurons, fast spiking interneurons (basket cells), and thalamic cells. The dynamics of basket cells, thalamic cells, and the somatic compartment of $L5_{PT}$ cells (Fig 1A-C) were described by Izhikevich quadratic adaptive integrate and fire neurons, a hybrid dynamical system that is

capable of reproducing a wide variety spiking behaviour while still being highly efficient to integrate numerically [34–36]. The Izhikevich neuron consists of the following two-dimensional system of ODEs:

$$C\dot{v}^{(s)} = k\left(v^{(s)} - v_r^{(s)}\right)\left(v^{(s)} - v_t^{(s)}\right) - u^{(s)} + I_{ext} \tag{1}$$

$$\dot{u}^{(s)} = a\left\{b\left(v^{(s)} - v_r^{(s)}\right) - u^{(s)}\right\} \tag{2}$$

with reset conditions: if $v \geq v_{peak}$ then $v \to c, u \to u + d$. The equations are in dimensional form giving the membrane potential (including the resting potential $v_r$, spike threshold $v_t$, and spike peak $v_{peak}$, and reset c), input $I_{ext}$, time $t$, and capacitance $C$, biophysically inter-pretable units (mV, pA, mS, and pF respectively). The remaining four parameters $k$, $a$, $b$, and d, are dimensionless and control the sharpness of the quadratic-nonlinearity, the timescale of spike adaptation, the sensitivity of spike adaptation to sub-threshold oscillations, and the magnitude of the spike reset adaptation variable. Crucially Izhikevich and colleagues [36,90], fit parameters for a large class of cortical and sub-cortical neurons, thus affording our model a high degree of neurobiological plausibility while greatly reducing the number of free parameters.

The apical compartment of the L5$_{PT}$ neuron consists of a two dimensional non-linear system introduced by [37] as a phenomenological model of the Ca$^{2+}$ plateau potential in the apical dendrites of L5$_{PT}$ neurons.

$$C\dot{v}^{(d)} = -l\left(v^{(d)} - v_r^{(d)}\right) + gf\left(v^{(d)}\right) + mH\left(t - t^s\right) + u^{(d)} + I_{ext} \tag{3}$$

$$\dot{u}^{(d)} = a\left\{b\left(v^{(d)} - v_r^{(d)}\right) - u^{(d)}\right\} \tag{4}$$

Where $f(x) = 1/(1 + \exp\left(-\dfrac{(x+38)}{6}\right)$ describes the regenerative non-linearity underlying the

Ca2+ plateau potential, and $H\left(t - t^s\right)$ denotes a square wave function of unitary amplitude describing the backpropagating action potential (delayed by 0.5 ms and lasts for 2 ms) with $t^s$ denoting the somatic compartment spike time. The parameters $l$, $g$, $m$, denote the leak conductance (nS), amplitude of regenerative non-linearity (pA), and amplitude of the back propagating action potentials (pA) respectively. The model and parameters were derived from a more complex model of Ca$^{2+}$ spikes built to predict *in vitro* L5$_{PT}$ spike times [42].

To simulate key observations from empirical experiments, we coupled the compartments together so that sodium spikes in the somatic compartment triggered a back propagating action potential affecting the apical compartment through the square wave function $H\left(t - t^s\right)$. In turn, plateau potentials in the apical compartment controlled the reset conditions of the somatic compartment. We leveraged the insight [34,40,41] that the difference between regular spiking and intrinsic bursting can be modelled by changing the reset conditions of equations [1] and [2], raising the reset voltage (increasing $c$) taking the neuron closer to threshold, and reducing the magnitude of spike adaptation (decreasing $d$). Whenever the membrane potential in the apical compartment exceeded $-30$ mv the reset conditions changed from regular spiking to bursting parameters. This allowed us to reproduce the transient change in dynamical regime in L5$_{PT}$ cells that occurs when they receive coincident apical and basal drive. Parameters values for each neuron/compartment are given in **Table 1**.

Based on the finding that communication between apical dendrites and the soma of L5$_{PT}$ cells requires depolarising input from the matrix thalamus to the "apical coupling zone" in L5a [43] we made back propagating action potentials and Ca²⁺ driven parameter switches depend stochastically upon a phenomenological model of excitatory dynamics in the apical coupling zone described by the saturating linear system shown in equation [5].

$$\dot{g}_{i,coupling} = -\frac{g_{i,coupling}}{\tau_{coupling}} + \left(1 - g_{i,coupling}\right)\sum_j \delta(t - t_j^s) \tag{5}$$

Coupling was driven by thalamic spikes (where $t^s$ denotes the time that the thalamic neuron passes the threshold $v \geq v_{peak}$) and the decay constant $\tau_{coupling}$ was taken from work estimating the decay of the post synaptic excitatory effects of metabotropic glutamate receptors [91] which have been shown empirically to mediate inter-compartmental coupling in L5$_{PT}$ cells [43]. By design, the dynamics of the coupling variable varied between 0 and 1 and governed the probability with which back propagating action potentials would reach the apical compartment and the probability with which a Ca²⁺ spike would lead to a switch in the soma reset parameters.

Based on previous spiking neural network models of rivalry [25,26,45] the cortical component of the network had a one-dimensional ring architecture. Each point on the ring represents an orientation preference with one full rotation around the ring corresponding to a 180° visual rotation. This mirrors the fact that a 180° rotation of a grating results in a visually identical stimulus and also ensures periodic boundary conditions. The cortical ring contained 90 L5$_{PT}$ neurons and 90 fast spiking interneurons. Each pair of excitatory and inhibitory neurons was assigned to an equidistant location on the ring (unit circle) giving each neuron a 2° difference in orientation preference relative to each of its neighbours. The (dimensionless) synaptic weights $w_{i,j}^{\kappa\omega}$ connecting neurons ($E \to E$, $I \to E$, and $E \to I$), were all-to-all with amplitude decaying as a function of the Euclidean distance $d_{i,j}$ between neurons (equation [6]) according to a spatial Gaussian footprint (equation [7]).

$$d_{i,j} = \sqrt{(\cos\theta_i - \cos\theta_j)^2 + (\sin\theta_i - \sin\theta_j)^2} \tag{6}$$

$$w_{i,j}^{\kappa\omega} = \lambda^{\kappa\omega} e^{-\frac{1}{2}\left(\frac{d_{i,j}}{\sigma^{\kappa\omega}}\right)^2} \tag{7}$$

Where $\theta$ is the location of the neuron on the unit circle, $\kappa$ and $\omega$ denote the pre- and post-synaptic neuron type (i.e. $E \to E$), $\lambda$ controls the magnitude of the synaptic weights,

**Table 1. Parameters for each neuron L5$_{PT}$ apical dendrite parameters were taken from [37]. L5$_{PT}$ soma parameters were modified from the model of intrinsic bursting described in [36], p.290). Basket cell (fast spiking interneuron), and matrix thalamus parameters were taken from [90].**

| Neuron | $C$ pF | $k$ a.u. | $v_r$ mV | $v_t$ mV | $a$ | $b$ | $c$ mV | $d$ a.u. | $v_{peak}$ mV | $l$ nS | $g$ pA | $m$ pA |
|---|---|---|---|---|---|---|---|---|---|---|---|---|
| L5$_{PT}$ apical dendrite | 170 | ~ | −70 | ~ | $\frac{1}{30}$ ms⁻¹ | −13 nS | ~ | ~ | ~ | 24.2857 | 1200 | 2600 |
| L5$_{PT}$ soma | 150 | 2.5 | −75 | −45 | 0.01 a.u | 5 a.u | RS: −65 IB: −55 | RS: 250 IB: 150 | 50 | ~ | ~ | ~ |
| Basket cell | 20 | 1 | −55 | −40 | 0.01 a.u | 8 a.u | −55 | 200 | 25 | ~ | ~ | ~ |
| Matrix thalamus | 200 | 1.6 | −60 | −50 | 0.01 a.u | 15 a.u | −60 | 10 | 35 | ~ | ~ | ~ |

and $\sigma^{\kappa\omega}$ the spatial spread. In line with empirical constraints inhibitory coupling had a larger spatial spread than excitatory to excitatory coupling [92]. Each thalamic neuron received input from 9 cortical neurons and then projected back up to the apical dendrites of the same 9 cortical neurons recapitulating the diffuse projections of higher-order thalamus onto the apical dendrites of $L5_{PT}$ neurons in layer 1 [47,48]. For simplicity we set projections to and from the thalamus to a constant value (e.g., $w_{i,j}^{TH\to D} = \lambda^{TH\to D}$). For the sake of computational efficiency we also neglected differences in rise time between receptor types which allowed us to model receptor dynamics with a first-order linear differential (equation [8]) with decay ($\tau_{decay}$) constants chosen to recapitulate the dynamics of inhibitory (GABA$_A$), and excitatory (AMPA and NMDA) synapses [93,94].

$$\dot{g}_{i,syn} = -\frac{g_{i,syn}}{\tau_{decay}} + w_{i,j}^{\kappa\omega}\sum_j \delta(t-t_j^s) \tag{8}$$

Where, as above, denotes $t^s$ the time that the neuron passes the threshold $v \geq v_{peak}$. The conductance term entered into the input $I_{ext}$ through the relation $I_{i,syn} = g_\infty g_i(t)_{syn}\left(E_{syn} - v_i(t)\right)$ where $E_{syn}$ is the reverse potential of the synapse. Following [90] we set $g_\infty$ to 1 for GABA$_A$ and AMPA synapses, and $g_\infty = \dfrac{\left[(v+80)/60\right]^2}{1+\left[(v+80)/60\right]^2}$ for NMDA synapses. To prevent artificial distortions of the spike shape that can occur during parameter sweeps that push the model outside its normal operating regime we clipped individual NMDA conductances to a maximum value of 85 nS.

For the threshold detection simulations, the somatic compartment of each $L5_{PT}$ cell received 600 Hz of independent (Poisson) external drive and apical compartments received 50 Hz of external drive. The whisker deflection was simulated by a pulse of constant amplitude varying between 0 – 350 pA lasting 200 ms and weighted by the spatial Gaussian shown in equation [9] where N is the neuron at the centre of the pulse and $\sigma^{TD}$ the spatial spread.

$$h_i^{TD} = e^{-\left(\frac{(i-N)}{\sigma^{TD}}\right)^2} \tag{9}$$

For the visual rivalry simulations, separate monocular inputs targeting the somatic compartment of $L5_{PT}$ cells were modelled with two independent Poisson processes (representing input from the left and right eyes in the case of binocular rivalry or left and right movement selective populations in the case of plaid perception) with rates varying between 1200 and 1800 Hz depending on the simulation. The external drive was weighted by the spatial Gaussian shown in equation [10] centred on neurons 90° apart on the ring abstractly corresponding to the orthogonal grating stimuli commonly employed in binocular rivalry experiments.

$$h_i^{VR} = e^{-\left(\frac{(i-N_L)}{\sigma^{VR}}\right)^2} + e^{-\left(\frac{(i-N_R)}{\sigma^{VR}}\right)^2} \tag{10}$$

Here $i$ denotes the index of the $i$ th neuron, $N_L$ and $N_R$ control the orientation of the stimulus delivered to the left and right eyes, and $\sigma^{VR}$ the spatial spread.

To capture the slow hyperpolarising current that traditionally governs switching dynamics in models of bistable perception [53], the somatic compartment of each $L5_{PT}$ cell was coupled to a phenomenological model of slow hyperpolarising Ca$^{2+}$ mediated K$^+$ currents [52] which entered into the external drive term for each cell (i.e. $I_{i,adapt} = g_i(t)_{adapt}\left(E_{adapt} - v_i(t)\right)$) with dynamics given by equation [10].

$$\dot{g}_{adapt} = -\frac{g(t)_{adapt}}{\tau_{adapt}} + \Delta g \delta(t - t^s) \tag{11}$$

Where $\Delta g$ denotes the contribution of each spike to the hyperpolarising current, and $\tau_{adapt}$ the decay constant.

Rather than fit the parameters of our model to individual experimental findings, which permits substantial degrees of freedom and risks overinterpretation of idiosyncratic aspects of individual experiments, we instead elected to challenge a single model to qualitatively reproduce a wide array of experimental findings with a minimal set of parameter changes carefully chosen to reflect experimental manipulations and perturbations. Specifically, we initialised the connectivity parameters such that: 1) when the model received a background drive the conductances were approximately E/I balanced with a coefficient of variation > 1, corresponding to an asynchronous irregular regime [49]; 2) inhibitory connections on the cortical ring had broader (Gaussian) connectivity than excitatory connections generating a winner-take-all regime when the model received two "competing" inputs to opposite sides of the cortical ring; and 3) a slow hyperpolarising current was added to the somatic compartment of each L5$_{PT}$ cell destabilizing the winner-take-all attractor states leading to spontaneous switches between transiently stable persistent states with an average duration in the experimentally observed range for binocular rivalry. Parameters for the model components described by equations [5] – [11] are supplied in **Table 2**.

The equations were integrated numerically in MATLAB 2023b. The apical compartment was integrated with a standard forward Euler scheme. All other compartments were integrated using the hybrid scheme for conductance based models introduced by [95]. All simulations used a step size of 0.1 ms and were run for 30 s. Unless stated otherwise, all simulation results were averaged over a minimum of 30 random seeds.

## Distance to bifurcation

To obtain a closed form expression for the distance to bifurcation in the apical compartment (equations [3-4]), we leveraged the fact that saddle node bifurcations occur when the nullclines ( $\dot{v}^{(d)} = \dot{u}^{(d)} = 0$ ) of the system intersect tangentially [58]. That is, the nullclines and derivative of the nullclines must be equal leading to the following two requirements (where we have absorbed the term describing back propagating action potentials $H(t - t^s)$ into the external drive $I_{ext}$ which we treat as a constant).

$$\frac{l}{C}\left(v^{(d)} - v_r^{(d)}\right) + \frac{1}{C}\left(gf\left(v^{(d)}\right) + I_{ext}\right) = b\left(v^{(d)} - v_r^{(d)}\right) \tag{12}$$

$$\frac{d}{dv}\left[\frac{l}{C}\left(v^{(d)} - v_r^{(d)}\right) + \frac{1}{C}\left(gf\left(v^{(d)}\right) + I_{ext}\right)\right] = \frac{d}{dv}\left[b\left(v^{(d)} - v_r^{(d)}\right)\right] \tag{13}$$

We used equation [13] to solve for $v^{(d)}$ giving $v^{(d)*} = \left[-44.6601, -31.3399\right]$. We then substituted $v^{(d)*}$ back into equation [12] to solve for $I_{ext}$ yielding the value of the external current at each of the two bifurcations I$_{B1}$ = 538.911 pA and I$_{B2}$ = 647.375 pA (corresponding to points at which the linear adaptation current nullcline intersects tangentially with the left and right knees of the cubic membrane potential nullcline see **Fig AB-AD in** S1 Appendix).

**Table 2. Parameter description, values and units for the model components described by equations [5–11].**

| Parameter | Description | Value | Units |
|---|---|---|---|
| $\lambda_{AMPA}^{E \to E}$ | Amplitude of excitatory to excitatory coupling for (AMPA) | $\dfrac{6.125}{\sigma^{E \to E}\sqrt{2\pi}}$ | a.u. |
| $\lambda_{NMDA}^{E \to E}$ | Amplitude of excitatory to excitatory coupling (NMDA) | $\dfrac{1.225}{\sigma^{E \to E}\sqrt{2\pi}}$ | a.u. |
| $\lambda^{E \to I}$ | Amplitude of excitatory to inhibitory coupling (NMDA and AMPA) | $\dfrac{1}{\sigma^{E \to I}\sqrt{2\pi}}$ | a.u. |
| $\lambda^{I \to E}$ | Amplitude of inhibitory to excitatory coupling (GABA$_A$) | $\dfrac{5}{\sigma^{I \to E}\sqrt{2\pi}}$ | a.u. |
| $\lambda^{E \to TH}$ | Constant excitatory to thalamic coupling constant (AMPA only) | 4 | a.u. |
| $\lambda_{AMPA}^{TH \to D}$ | Constant thalamic to apical dendrite coupling (AMPA) | 10 | a.u. |
| $\lambda_{NMDA}^{TH \to D}$ | Constant thalamic to apical dendrite coupling (NMDA) | 10 | a.u. |
| $\sigma^{E \to E}$ | Spread of excitatory to excitatory coupling | 0.5 | a.u. |
| $\sigma^{E \to I}$ | Spread of excitatory to inhibitory coupling | 2 | a.u. |
| $\sigma^{I \to E}$ | Spread of inhibitory to excitatory coupling | 2 | a.u. |
| $\tau_{decay}$ : AMPA | Decay time of AMPA conductance | 6 | ms |
| $\tau_{decay}$ : GABA$_A$ | Decay time of GABA$_A$ conductance | 6 | ms |
| $\tau_{decay}$ : NMDA | Decay time of NMDA conductance | 100 | ms |
| $\tau_{coupling}$ | Decay time of apical coupling zone | 800 | ms |
| $\tau_{adapt}$ | Decay time of adaptation current | 2000 | ms |
| $\Delta g$ | Contribution of each spike to adaptation current | 0.065 | nS |
| $\sigma^{TD}$ | Spatial spread of external drive | 20 | a.u. |
| $\sigma^{VR}$ | Spatial spread of apical drive | 18 | a.u. |
| $E_{Excitatory}$ | Reverse potential of excitatory synapses | 0 | mV |
| $E_{Inhibitory}$ | Reverse potential of inhibitory synapses | -75 | mV |
| $E_{Adapt}$ | Reverse potential of adaptation currents. | -80 | mV |

## Psychometric and neurometric functions

To obtain a measure of response probability from our model comparable to the psychometric functions in Takahashi and colleagues [12,13] we took a two pronged approach. First, for all stimulus intensities including stimulus absent trials (when the model only received a background drive) we calculated the frequency with which the spike count in the 1000 ms post stimulus window exceeded a criterion defined on the interval between the minium and maximum spike count. We then selected the (optimal) criterion that best minimised misses and false alarms. Trials exceeding the optimal criterion were counted as a response. Following Takahashi and colleagues [12,13], we then fit logistic functions (equation [14]) to the network responses using non-linear least-squares.

$$P\left(x;\alpha,\beta,\lambda,\gamma\right)=\gamma+\frac{1-\gamma-\lambda}{1+e^{-\beta(x-\alpha)}} \tag{14}$$

Where *P(x)* is the detection probability (i.e. the probability of the model producing a hit or false alarm), and $\alpha,\beta,\lambda,\gamma$ are free parameters. We used the optimal criterion found in the unperturbed (i.e. control) simulations in the perturbation simulations.

Second, to ensure that our results were not an artefact of the (optimal) criterion we constructed neurometric functions following the procedure described in the supplementary material of (13). Specifically, for each stimulus intensity we constructed ROC curves and then computed the AUC (area under the ROC curve) thereby summing over all criterions. To convert the AUC into a quantity comparable to a psychometric function (i.e. so that each neurometric function vairied between 0 and 1), we normalised the AUC values, $P(response) = (AUC - intercept) \, max$. The *intercept* was given by the minimium AUC across all conditions, and the *max* was given by the maximum AUC across all conditions.

## Supporting information

**S1 Appendix.**   ***Section A. Apical compartment phase plane.*** **Fig A. A)** Bifurcation diagram of the $L5_{PT}$ apical compartment. The saddle node bifurcation at $I_{B1}$ generates a stable plateau potential which coexists with the resting state of the apical compartment until the model passes through a second saddle node bifurcation at $I_{B2}$ at which point the resting state of the compartment vanishes and the plateau potential becomes globally attracting. **B-C)** Phase plane representation of the apical compartment showing the nullclines (black) for the following values of the bifurcation parameter; $I_{ext} < I_{B1}$, $IB1 < I_{ext} < I_{B2}$, $I_{ext} > I_{B2}$. ***Section B. Sweeping the magnitude of model perturbations.*** **Fig B. A-C)**Psychometric function fit to spiking model output across apical compartment excitation (blue), apical compartment inhibition (pink), and thalamic inhibition (orange), of varying magnitudes; **A** = 300 pA, **B** = 200 pA, **C** = 100 pA. **D-F)** Same as A-C but for neurometric functions; **D** = 300 pA, **E** = 200 pA, **F** = 100 pA.***Section C. Robustness of rivalry duration across burstiness parameters.*** **Fig C.**Mean dominance duration as a function of the spike reset parameter values, and inter-compartment coupling probability, controlling the burstiness of the model $L5_{PT}$ cells. **A)** Dominance duration as a function of spike reset values with dynamic inter-compartment coupling (typical values are ~ 0.98 for the dominant population, and ~ 0.1 for the suppressed population; see **Fig 3D**). **B – F)** Dominance duration as a function of spike reset values with stationary coupling probability ranging from 1 to 0.6.***Section D. Dynamical regime underlying visual rivalry.*** **Fig D. A)** Average firing rate of neuronal populations centred on opposite ends of the ring driven by a constant drive with asymmetric initial conditions. **B)** Average firing rate of neuronal populations simulated with constant drive and symmetric initial conditions. A perturbation was delivered at $t = 5000$ ms sending the population orbit to the surrounding stable limit cycle. **C)** Average firing rate of neuronal populations driven by constant drive with asymmetric initial conditions. Perturbation delivered at $t = 5000$ ms had no substantial effect on the already oscillating orbit indicative of a stable limit cycle. ***Section E. Key effects of visual rivalry simulations are preserved in scaled-up model.*** **Table A.** Parameter description, values, and units of the (scaled-up) model components described by equations [5-11]. **Fig E. A)** Raster plots of somatic spikes from the scaled-up population of $L5_{PT}$ cells **B)** Histogram of dominance durations, black line shows the fit of a Gamma distribution with parameters estimated via MLE ($\alpha$ = 6.2, $\theta$ = 0.56). **C)** Simulation confirming Levelt's second proposition in scaled-up model. Dashed line shows the dominance duration of the population receiving the decreasing external drive, solid line shows dominance duration of population receiving a fixed drive. **D)** Simulation of Levelt's fourth proposition in scaled-up model. ***Section F. In silico electrophysiology supplemental figures.*** **Fig F. A)** Inter-compartment coupling probability under asymmetric perturbation for perturbed (solid) and unperturbed (dashed) populations as a function of perturbation strength for each perturbation type (colours same as main text). **B)** Dominance duration standard deviation under asymmetric perturbation as a function of the strength of each perturbation type. **C)** Inter-compartment coupling probability under symmetric perturbations. **D)** Dominance

duration standard deviation under symmetric perturbation as a function of perturbation strength for each perturbation type.
(PDF)

## Acknowledgments

We are grateful to Michael Breakspear and Bryan Paton for insightful discussion on the initial development of the model. The Sydney Systems Neuroscience and Complexity group, David Alais, Matt Davidson, Jacob Coorey, and Hugh Wilson provided discerning comments and advice that helped to shape the presentation and structure of the manuscript.

## Author contributions

**Conceptualization:** Christopher J. Whyte, Eli J. Müller, Jaan Aru, Matthew Larkum, Yohan John, Brandon R. Munn, James M. Shine.

**Data curation:** Christopher J. Whyte.

**Formal analysis:** Christopher J. Whyte, Eli J. Müller, Brandon R. Munn, James M. Shine.

**Funding acquisition:** James M. Shine.

**Investigation:** Christopher J. Whyte, Brandon R. Munn, James M. Shine.

**Methodology:** Christopher J. Whyte, Eli J. Müller, Brandon R. Munn, James M. Shine.

**Software:** Christopher J. Whyte, Brandon R. Munn.

**Supervision:** Eli J. Müller, Brandon R. Munn.

**Visualization:** Christopher J. Whyte.

**Writing – original draft:** Christopher J. Whyte.

**Writing – review & editing:** Christopher J. Whyte, Eli J. Müller, Jaan Aru, Matthew Larkum, Yohan John, Brandon R. Munn, James M. Shine.

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
