## [Decision Letter · Decision Letter 0]

22 Oct 2024

Dear Mr Whyte,

Thank you very much for submitting your manuscript "Burst-dependent Thalamocortical Dynamics Underlie Perceptual Awareness" for consideration at PLOS Computational Biology.

As with all papers reviewed by the journal, your manuscript was reviewed by members of the editorial board and by several independent reviewers. In light of the reviews (below this email), we would like to invite the resubmission of a significantly-revised version that takes into account the reviewers' comments.

We cannot make any decision about publication until we have seen the revised manuscript and your response to the reviewers' comments. Your revised manuscript is also likely to be sent to reviewers for further evaluation.

Sincerely,

Boris S. Gutkin

Academic Editor

PLOS Computational Biology

Hugues Berry

Section Editor

PLOS Computational Biology

Reviewer's Responses to Questions

**Comments to the Authors:**

Reviewer #1: The authors develop a semi-biophysical model for the interactions between layer 5 pyramidal neurons, inhibitory interneurons, and cells in the matrix thalamus. I cannot speak for the realism of the connectivity (I could find little in the literature about the matrix thalamus that was not written by one of the authors of this paper; the little I did find suggested that the calbindin neurons (matrix) projected mostly to the superficial layers of the ctx, I, II)

The paper seems more like two papers. One would be very short and concerns the responses of layer V neurons in the barrel cortex of mice to whisker flicks and how they are modulated by optogenetic and pharmacological manipulation. Again, the circuitry here surprised me as my understanding was that the flick of the whisker activated VPM which then feeds directly forward to layer IV of the barrel cortex, so i wonder what happened to that part of the circuit. I also thought layer IV -> II/III which interacts with V, so how does one eliminate that major circuit?

The part of the paper on rivalry (BR) doesnt really distinguish itself from other models although, in its favor, it provides some experimental predictions. However, I would like to see more details on what BR experiments can be done on mice. Plaids are mentioned, but no details are given. I really dont see any easy way to tie these two parts of tyhe paper together. The ring structure seems to be irrelevent to the whisker flicking and pretty much any network with a ring structure and long range inhibition will give rise to WTA type dynamics. Since the Py cells have adaptation, this is basically the same model as others have proposed. The authors make a big deal about the importance of the bursting properties of the Py cells but it is unclear why that is important for BR. I dont know much about the thalamic matrix, but in the other sensory thalami, the TC cells have a prominent T-type calcium current and thus are capable (and in fact do so routinely) of producing bursts. They are also subject to modulation that takes them from burst to relay mode; there is a vast and old literature on this as well as models going back at least to the 90's. There are also reticular inhibitory neurons which seem to be missing in the model. Perhaps they are not in the matrix thalamus.

Other points:

page 9 - is there a reason the thalamic projection dont synapse the inhibitory neurons? That is certainly the case in layer IV

in Fig 1B, the firing patterns of the neurons -- are they all synaptially coupled in the network? Why does one excitatory spike lead to a long burst of activity in the inhibitory neuron? Is there some long lasting synapse?

page 12

The data in fig 2c seem to show a much sharper threshold than the model in the control case. Is there a reason for this? Why are the data controls in c and d so completely different? The thresholds are also very different. Not sure how much credence to put in the data if the controls are so different despite being under identical circumstances

page 14

Distance to bifurcation seems to be a major part of the paper. How did you compute this distance from the fold? The model is stochastic so it is not clear to me how you get such precision in distance metrics. I understand how you find the folds as that is just algebra, but how do you get the values in fig 2I

How was pharmacological inhibition mediated in the model? How do you know how strong to make it? Do you just change it until the model matches the data?

page 16

In 3 D, what are the black curves?

page 17

The whole bit about analytic proofs of limit cycles is bizarre and unnecessary. This is not really relevant as your odes are intrinsically stochastic in that there is a probability of switching to bursting, and it is not my impression that you turned that “noise” off. In a purely deterministic model, the hopf theorem provides a way to prove existence of limit cycles in any dimension. This whole section should probably be rewritten or eliminated

page 18

What happens with continued lowering of the input strengths? In the models that were collected in Shpiro et al, they find that the period is non monotone with respect to strength. I’d be interested to see what happens as the stimulus decreases. In most cases there are hopf bifurcations at each end.

page 20

What does perturbation strength mean for unperturbed conditions? Why isn’t it flat ?

page 30

you use f(x) and f(s) for different things. Change symbols

Reviewer #2: This manuscript presents a computational model of thalamocortical networks which is able to explain recent experimental findings regarding the role of dendro-somatic communication of layer 5 pyramidal cells in perceptual awareness. The authors then take their biophysically realistic model and use it to explore experimental findings on visual rivalry, thus extending the original claims of the experimental papers towards other tasks and sensory modalities, replicating classical results such as Levelt’s binocular rivalry propositions, and providing new testable predictions.

The topic is timely and relevant, the work is carried out diligently, and the results will certainly be of interest for the computational neuroscience community, as well as for researchers in perception and awareness topics. The current version however contains a few limitations and modeling choices that might need to be reassessed in order to be fully convincing. These issues are summarized below.

1) A network of 180 neurons is definitely on the lower side in terms of network size. Spiking networks that small might suffer from finite size effects and provide misleading results. This is even worse for the present case, as the orientation selectivity ring structure means that each window of 2 degrees will only have one excitatory neuron representing it. In real neural circuits, we would expect to have at least a small population for each (approximate) orientation. Given that the authors use Izhikevich models for their neurons, which is a highly efficient model computationally, it should not we a problem to go into a more reasonable size of 5,000-10,000 neurons, so that each box of 2 degrees is encoded by approximately 50-100 neurons.

2) The cortical ring seems to have an equal number (90) of excitatory and inhibitory cells. This is a strange choice given that the E:I ratio in most cortical regions is about 80:20. A larger number of excitatory cells, especially when recurrently connected (see point below) could lead to destabilizing the system’s activity or change the bursting patterns, and therefore it’s important that the authors show whether their main findings hold for this more realistic ratio.

3) Related to the above two points, simulating a larger network would permit the existence of recurrent connectivity between each pool of 50-100 L5 pyramidal cells encoding the same orientation, which is something expected from a realistic model. The absence of recurrent connectivity could hinder potential problems with the model, such as pathological synchrony or hyperexcitability. A supplementary study showing the impact of such recurrent connections could be done to dissipate doubts.

4) The model keeps the same parameters to simulate the tactile threshold detection as well as visual rivalry tasks. While I appreciate and agree with the idea that showing a generic model perform these tasks is more convincing than having individually tuned models, it is also true that circuit properties might substantially vary across species. Cortical pyramidal cells, in particular, are known to display substantial differences between mice and primates (Gilman et al. Cereb. Cortex 2017, Mihaljevic et al. Front. Neuroinform. 2021, Kalmbach et al. Neuron 2018 & 2021). Therefore, it is possible that some of the results might be substantially vary across species and the model might be missing this fact by committing to only a given parameter set. One way to solve this would be to show how robust the main results are when certain parameters are varied in the model, to account for species-related differences in cell physiology. This could be shown in a supplementary figure, for example.

5) Fig 2 seems to show that model predictions regarding the upwards shift of the psychometric functions tend to underestimate the magnitude of the shift for strong inputs. Would this be a limitation from the data (i.e. experiments not going for strong enough inputs) or from the model (i.e. absence of adaptation mechanisms for strong input)? A brief discussion about this somewhere in the text would be useful.

6) It might be interesting to test whether the visual rivalry results change in the model for stimuli not fully orthogonal. Is the statistics of dominance durations maintained? At which point (i.e. proximity between both stimuli) the bistable competition breaks down? This could provide exciting predictions for future psychophysics experiments, and would be easy to test in the model.

**Have the authors made all data and (if applicable) computational code underlying the findings in their manuscript fully available?**

Reviewer #1: Yes

Reviewer #2: **No: ** The authors have provided part of the code, the rest will be available upon acceptance of the manuscript.

PLOS authors have the option to publish the peer review history of their article (what does this mean? ). If published, this will include your full peer review and any attached files.

**Do you want your identity to be public for this peer review?** For information about this choice, including consent withdrawal, please see our Privacy Policy .

Reviewer #1: No

Reviewer #2: No
---

## [Decision Letter · Decision Letter 1]

24 Feb 2025

PCOMPBIOL-D-24-01251R1

A Burst-dependent Thalamocortical Substrate for Perceptual Awareness

PLOS Computational Biology

Dear Dr. Whyte,

Thank you for submitting your manuscript to PLOS Computational Biology. After careful consideration, we feel that it has merit but does not fully meet PLOS Computational Biology's publication criteria as it currently stands. Therefore, we invite you to submit a revised version of the manuscript that addresses the points raised during the review process.

Please submit your revised manuscript within 30 days Apr 26 2025 11:59PM. If you will need more time than this to complete your revisions, please reply to this message or contact the journal office at ploscompbiol@plos.org. Please include the following items when submitting your revised manuscript:

We look forward to receiving your revised manuscript.

Kind regards,

Boris S. Gutkin

Academic Editor

PLOS Computational Biology

Hugues Berry

Section Editor

PLOS Computational Biology

**Journal Requirements:**

1) Please provide an Author Summary. This should appear in your manuscript between the Abstract (if applicable) and the Introduction, and should be 150-200 words long. The aim should be to make your findings accessible to a wide audience that includes both scientists and non-scientists. Sample summaries can be found on our website under Submission Guidelines:

2) We notice that your supplementary Figures, and Table are included in the manuscript file. Please remove them and upload them with the file type 'Supporting Information'. Please ensure that each Supporting Information file has a legend listed in the main file of the manuscript after the references list.

Note: The Supporting Information legends should be included in the main file of the manuscript after the references list. In addition, the Supporting files (figures, table, information) should be uploaded separately with the file type 'Supporting Information'.

**Reviewers' comments:**

Reviewer's Responses to Questions

Reviewer #1: The authors have substantially improved the paper and I am happy with their changes and now buy into the combination of the two parts. I have only a few remarks:

Regarding Fig 3F

I must be misreading the figure. If I look at the mean values, the solid line goes from 3.5 to 5 and the dashed from 3.5 to 1.5, so that this in fact contradicts the second principle as (5-3.5) < (3.5-1.5)

Of course if you take the highest value of the std dev bar for the solid and the highest for the dashed, the you obtain what the authors say here, but that seems a little unfair and the means tell a quite different story

The authors emphasize the burst dependent mechanisms in perception. So, I think they should run some rivalry simulations when they reduce the bursting for example, by gradually decoupling the soma and dendrite. Most other models of BR dont need bursting so I remain skeptical of why this is necessary. Some of this is sort of done in Fig 5, but I'd like a more systematic study

Please replace "whilst" with "while"

Reviewer #2: The authors have satisfactorily addressed my concerns, including a verification of their main results with a scaled-up model and preliminary results related to one of my questions (but which will be published elsewhere). I consider that the manuscript is now in great shape, and ready to be published.

Jorge Mejias

**Have the authors made all data and (if applicable) computational code underlying the findings in their manuscript fully available?**

Reviewer #1: Yes

Reviewer #2: Yes

PLOS authors have the option to publish the peer review history of their article (what does this mean? ). If published, this will include your full peer review and any attached files.

**Do you want your identity to be public for this peer review?** For information about this choice, including consent withdrawal, please see our Privacy Policy .

Reviewer #1: No

Reviewer #2: **Yes: ** Jorge Mejias

**Figure resubmission:**
---

## [Editor Report · Decision Letter 2]

11 Mar 2025

Dear Mr Whyte,

We are pleased to inform you that your manuscript 'A Burst-dependent Thalamocortical Substrate for Perceptual Awareness' has been provisionally accepted for publication in PLOS Computational Biology.

Best regards,

Boris S. Gutkin

Academic Editor

PLOS Computational Biology

Hugues Berry

Section Editor

PLOS Computational Biology

---

## [Editor Report · Acceptance letter]

PCOMPBIOL-D-24-01251R2

A Burst-dependent Thalamocortical Substrate for Perceptual Awareness

Dear Dr Whyte,

I am pleased to inform you that your manuscript has been formally accepted for publication in PLOS Computational Biology. Your manuscript is now with our production department and you will be notified of the publication date in due course.

With kind regards,

Anita Estes
